# SIMULATING, FAST AND SLOW: LEARNING POLICIES FOR BLACK-BOX OPTIMIZATION

## ABSTRACT

Simulators are vital in science and engineering, as they faithfully model the influence of design parameters on real-world observations. A common problem is leveraging the simulator to optimize the design parameters to minimize a desired objective function. Since simulators are often non-differentiable blackboxes and each simulation incurs significant compute time, gradient-based optimization techniques can often be intractable or, in some cases, impossible. Furthermore, in many experiment design settings, practitioners are required to solve sets of closely related optimization problems. Thus, starting the optimization from scratch each time might be inefficient if the forward simulation model is expensive to evaluate. To address these challenges, this paper introduces a novel method for solving classes of similar black-box optimization problems by learning an active learning policy that guides the training of a differentiable surrogate and then uses that surrogate's gradients to optimize the simulation parameters with gradient descent. After training the policy, the cost for downstream optimization of problems involving black-box simulators is amortized and we require up to ∼90% fewer expensive simulator calls compared to baselines such as local surrogate-based approaches, numerical optimization, and Bayesian methods.

## 1 INTRODUCTION

Simulation-based techniques model real-world phenomenons (e.g., physics particle movement, electromagnetic wave propagation) and enable understanding influence of system design parameters on resulting observations. As such, they provide a cheaper alternative to real-world evaluation of system parameters and are invaluable to many fields in physical sciences and engineering, covering domains such as robotics (Todorov et al., 2012), telecommunication (Hoydis et al., 2023) and particle physics (Jonas, 2019; Stakia, 2021). Generally, a simulator[1] $f_{\text{sim}}$ models the *forward*-process $f_{\text{sim}} : (\boldsymbol{\psi}, \boldsymbol{x}) \to \boldsymbol{y}$, which maps simulation parameters $\boldsymbol{\psi}$ and input data $\boldsymbol{x}$ to observations $\boldsymbol{y}$ (Shirobokov et al., 2020). For instance, in particle physics, simulators such as GEANT4 (Agostinelli et al., 2003) or FairRoot (Al-Turany et al., 2012), predict the detection of particles $\boldsymbol{y}$ given their properties $\boldsymbol{x}$, and multi-stage steel magnet configuration and geometry $\boldsymbol{\psi}$. Similarly, in wireless communication, simulators such as Matlab RT (Inc., 2023) or Sionna (Hoydis et al., 2023), predict the signal strength $\boldsymbol{y}$ given scene information $\boldsymbol{\psi}$ (e.g., CAD model of scene, antenna locations and orientations).

Although simulators largely focus on highly-accurate *forward*-models, numerous practical applications require *inverse* inferences. Specifically, inferring unknown system design parameters $\boldsymbol{\psi}^*$ that achieves a certain objective. Continuing the previous examples, in the particle physics scenario, to design the magnet configuration to reduce the number of detected events from certain types of particles. Similarly, in the wireless communication, to optimally place a transmit antenna in a scene to maximize the signal strength across all areas. Tackling inverse problems using simulators-in-the-loop can be cast as a *black-box optimization problem*: to iteratively refine an initial design parameter choice to meet the objective given certain conditions and constraints. Black-box optimization has rich history, and solutions include gradient-free optimization (Banzhaf et al., 1998; Maheswaranathan et al., 2019), Bayesian optimization (Daxberger et al., 2020; Oh et al., 2018), numerical differentiation

---

[1]In our study, we consider stochastic and non-stochastic simulators. Our method applies to both types of simulators without requiring any modifications.

**Figure 1: Schematic view of our approach**. (a) We study black-box optimization problem (over parameters $\psi$), with an emphasis on using gradient information from a fast differentiable surrogate $f_\phi$ (b) To optimize $\psi$ sample-efficiently, we employ a policy $\pi_\theta$ to actively determine whether retraining the surrogate is necessary before using the gradient information.

(Alarie et al., 2021; Shi et al., 2023) or stochastic gradient estimation methods (Grathwohl et al., 2018; Williams, 1992). However unlike typical blackbox settings, a major challenge here is that each simulation (for a fixed choice of $\psi$) involve significant compute and hence posing a critical bottleneck for iterative optimization. Consequently, we focus on blackbox optimization techniques that minimize calls to the simulator.

In this paper, we focus on stochastic gradient estimation techniques for black-box optimization. Inspired by Shirobokov et al. (2020), our approach involves leveraging gradients from a surrogate model trained to (locally) mimic the black-box simulator[2]. Gradient-based methods typically perform multiple simulator calls to estimate the gradients, thus making these approaches computationally demanding. To mitigate such a demand, we aim to minimize the number of required simulator calls by proposing to learn a *policy* to guide the optimization. The policy determines whether the current surrogate model (fast, but potentially inaccurate) can be used or instead a simulator call is necessary to update the surrogate (slow, but accurate), see Figure 1. Furthermore, by drawing inspiration from the literature on active learning (Bakker et al., 2023; Fang et al., 2017; Hsu and Lin, 2015; Konyushkova et al., 2017; 2018; Liu et al., 2018; Pang et al., 2018; Ravi and Larochelle, 2018) we also let our policy learn how to sample new data for training the local surrogate model. This offers additional control, which the policy may learn to exploit.

Our contribution can be summarized as follows: (i) We introduce a Reinforcement Learning (RL) framework to learn a policy to reduce the number of computationally expensive calls to a black-box simulator required to solve an optimization problem; (ii) We propose to learn a policy that determines when a simulator call is necessary to update the surrogate and when the current surrogate model can be used instead; (iii) We implement a policy that also learns how to sample new data for training the surrogate model during the optimization process; (iv) We assess the benefits of our RL-based approach on low- and high-dimensional global optimization benchmark functions and two real-world black-box simulators and show that, once trained, our policy reduces the number of simulator calls up to $\sim$90%, compared to the baselines.

## 2 RELATED WORK

**Simulation-based Inference** Our work lies at the intersection of black-box simulator-based optimization and active learning. Black-box optimization problems are ubiquitous in science and engineering, encompassing scenarios where unknown parameters must be deduced from observational data. These parameters can entail anatomies in MRI (Zbontar et al., 2018), molecular structures (Jonas, 2019), particle properties (Agostinelli et al., 2003; Stakia, 2021), and cosmological model parameters (Cole et al., 2022), among others. The forward process is often a complex physical process that can be modelled by a simulator but does not provide a likelihood for easy inference. Simulation-based inference techniques aim to infer posterior distributions over these simulation parameters in such likelihood-free settings (Brookes et al., 2019; Cole et al., 2022; Cranmer et al., 2020). Other solutions may involve supervised learning on observation-parameter pairs or imitation learning (Jonas, 2019; Sriram et al., 2020). Our simulator-based optimization setting is a variation on these problems. Here, the objective is to find the optimal parameters of the simulator, where optimality is typically formulated in terms of desired observations. This methodology can be applied in various fields, such as MRI (Bakker et al., 2022; 2020; Pineda et al., 2020), particle physics (Dorigo

---

[2]The simulator can be either stochastic or deterministic.

et al., 2023; Fanelli, 2022; Gorordo et al., 2023; Stakia, 2021) and molecular design (Schwalbe-Koda et al., 2021). When the simulators are differentiable, direct gradient-based optimization can perform well (de Avila Belbute-Peres et al., 2018; Degrave et al., 2019; Hu et al., 2019). However, a different approach is necessary in cases where the simulators are non-differentiable. Well-known gradient-free methods that may be employed in such settings include evolutionary strategies (Banzhaf et al., 1998; Maheswaranathan et al., 2019) and Bayesian optimization (Daxberger et al., 2020; Eriksson et al., 2019; Frazier, 2018; Oh et al., 2018). Nevertheless, these methods often require additional assumptions to make the optimization scalable in high dimensional parameter spaces (Djolonga et al., 2013; Zhang et al., 2019).

**Approximate-Gradient Optimization** With the rise of deep learning, there has been a surge of interest in approximate-gradient optimization methods. While some authors consider numerical differentiation (Alarie et al., 2021; Shi et al., 2023), many others have focused on methods for efficiently obtaining approximate stochastic gradients (Agrawal et al., 2023; Grathwohl et al., 2018; Louppe et al., 2019; Mohamed et al., 2020; Ruiz et al., 2019; Williams, 1992). Another strategy involves training differentiable surrogate models to mimic the simulator and assuming that the gradients of the surrogate model are similar enough to those of the simulator (Shirobokov et al., 2020). Surrogate models have been trained for many applications, including wireless propagation modeling (Levie et al., 2021; Orekondy et al., 2023), space weather prediction (Baydin et al., 2023), material discovery (Merchant et al., 2023), and fluid dynamics simulation (Agrawal and Koutsourelakis, 2024). This trend provides an opportunity for surrogate-based optimization of simulators, as surrogate models are readily available. Additionally, it has been observed by Shirobokov et al. (2020) that using (local) surrogate gradients is more efficient than many alternatives. Our work generalizes this setup by introducing a policy that guides the optimization by suggesting when and, optionally, how the surrogate should be updated during the optimization process.

**Active Learning** When the policy decides how (with what data) the surrogate should be updated, it does so using information provided by the surrogate itself. This is an example of active learning (Settles, 2009), where the current instance of a task model (the surrogate) affects the data it sees in future training iterations. In particular, our policies are instances of *learning active learning*, where a separate model (our policy) is trained to suggest the data that the task model should be trained on (Bakker et al., 2023; Fang et al., 2017; Hsu and Lin, 2015; Konyushkova et al., 2017; 2018; Liu et al., 2018; Pang et al., 2018; Ravi and Larochelle, 2018).

## 3 BACKGROUND

We aim to optimize the simulation parameters of a black-box simulator using stochastic gradient descent. The black-box simulator, $f_{\text{sim}}$, describes a stochastic process[3], $p(\boldsymbol{y}|\boldsymbol{\psi}, \boldsymbol{x})$, from which we obtain the observations as $\boldsymbol{y} = f_{\text{sim}}(\boldsymbol{\psi}, \boldsymbol{x}) \sim p(\boldsymbol{y}|\boldsymbol{\psi}, \boldsymbol{x})$, where $\boldsymbol{x} \sim q(\boldsymbol{x})$ is a stochastic input and $\boldsymbol{\psi}$ is the vector of simulation parameters. Since these simulators are typically not differentiable, we train a surrogate neural network to locally (in $\boldsymbol{\psi}$) approximate the simulator (Shirobokov et al., 2020). Gradients of these local surrogates, obtained through automatic differentiation, may then be used to perform the optimization over $\boldsymbol{\psi}$. The goal is now to minimize an expected loss $\mathcal{L}$ over the space of the simulation parameters $\boldsymbol{\psi}$. As the functional form of the simulator is generally unknown, this expectation cannot be evaluated exactly and is instead estimated using $N$ Monte Carlo samples:

$$\boldsymbol{\psi}^* = \arg\min_{\boldsymbol{\psi}} \mathbb{E}\left[\mathcal{L}(\boldsymbol{y})\right] = \arg\min_{\boldsymbol{\psi}} \int \mathcal{L}(\boldsymbol{y}) \, p(\boldsymbol{y}|\boldsymbol{\psi}, \boldsymbol{x}) \, q(\boldsymbol{x}) \, d\boldsymbol{x} \, d\boldsymbol{y} \approx \arg\min_{\boldsymbol{\psi}} \frac{1}{N} \sum_{i=1}^{N} \mathcal{L}(f_{\text{sim}}(\boldsymbol{\psi}, \boldsymbol{x}_i))$$

(1)

After training a neural network surrogate $f_{\phi} : (\boldsymbol{\psi}, \boldsymbol{x}, \boldsymbol{z}) \rightarrow \boldsymbol{y}$ on data generated with $f_{\text{sim}}$, the optimization might be performed following gradients of the surrogate. Here, $\boldsymbol{z}$ is a randomly sampled latent variable that accounts for the stochasticity of the simulator. Gradients are then estimated as: $\nabla_{\boldsymbol{\psi}} \mathbb{E}\left[\mathcal{L}(\boldsymbol{y})\right] \approx \frac{1}{N} \sum_{i=1}^{N} \nabla_{\boldsymbol{\psi}} \mathcal{L}\left(f_{\phi}(\boldsymbol{\psi}, \boldsymbol{x}_i, \boldsymbol{z}_i)\right)$. Since running the forward process $f_{\text{sim}}$, is often an expensive procedure, our goal is to minimize the number of simulator calls required to solve the optimization problem at hand.

---

[3] A non-stochastic simulator can be considered as a special case where $f_{\text{sim}}$ places a delta distribution over observations.

## 4 POLICY-BASED BLACK-BOX OPTIMIZATION

Following Shirobokov et al. (2020), we perform an iterative optimization based on the gradients obtained in section 3. At each point during the optimization, new values $\boldsymbol{\psi}_j$ are sampled within a box of fixed size $2\epsilon$, centered around the current $\boldsymbol{\psi}$: $U_\epsilon^{\boldsymbol{\psi}} = \{\boldsymbol{\psi}'; |\boldsymbol{\psi}' - \boldsymbol{\psi}| \leq \epsilon\}$. Then, input samples are obtained from $q(\boldsymbol{x})$, and the simulator is called to obtain the corresponding $\boldsymbol{y}$ values. The resulting samples are stored in a history buffer $H$, from which the surrogate is trained from scratch. Specifically, the surrogate is trained on samples $\boldsymbol{\psi}_j$ extracted from $H$ that satisfy the condition that they lie within $U_\epsilon^{\boldsymbol{\psi}}$. The overall process required to generate new samples from the black-box simulator is what we refer to as a "*simulator call*".

**Policy-based Approach**   We propose further reducing the number of simulator calls required for an optimization run with an RL-based approach. Our method involves utilizing a learned policy $\pi_\theta$, with learnable parameters $\theta$ to: i) decide whether a simulator call should be performed to retrain the local surrogate; and ii) define how to sample from the black-box simulator.

**Sampling Strategy**   To investigate the question concerning *how* to perform a simulator call, we train policies to additionally output the $\epsilon$ for constructing the sampling neighbour $U_\epsilon^{\boldsymbol{\psi}}$, which serves as our data acquisition function. As $\epsilon$ parameterizes this acquisition function, such policies are an example of active learning (Settles, 2009). In particular, these policies are instances of learning active learning (Bakker et al., 2023; Fang et al., 2017; Hsu and Lin, 2015; Konyushkova et al., 2017; 2018; Liu et al., 2018; Pang et al., 2018; Ravi and Larochelle, 2018), as they learn a distribution over $\epsilon$. See Appendices B.1 and C.1 for a more detailed description concerning the policy implementation and training.

**State Definition**   We formalize the sequential optimization process as an episodic Markov Decision Process (MDP). The state $s_t$ (at timestep $t$) is given by the tuple $(\boldsymbol{\psi}_t, t, l_t, \sigma_t)$, where $\boldsymbol{\psi}_t$ is the current parameter value, $l_t$ is the number of simulator calls already performed in the episode, and $\sigma_t$ is some measure of uncertainty produced by the surrogate. See Appendix B.3 for a discussion regarding observability in the MDP.

**Action Definition**   Actions $a_t$ consist of binary valued variables $b \in \{0, 1\}$, sampled from a Bernoulli distribution, where $1$ represents the decision to perform a simulator call. Additionally, we also train policies to determine, as part of the action, the trust region size $\epsilon$ for sampling new values for $\boldsymbol{\psi}$. The dynamics of the MDP is represented by means of the Adam optimizer (Kingma and Ba, 2014) which updates the current state by performing a single optimization step in the direction of the gradients of $\boldsymbol{\psi}$.

**Reward Design**   Episodes come to an end under three conditions: $\mathcal{A}$) when the optimization reaches a parameter for which $\mathbb{E}\left[\mathcal{L}(\boldsymbol{y})\right]$ is below the target value $\tau$ (we call this *termination* - see Appendix D.4 for details concerning the choice of $\tau$); $\mathcal{B}$) when the maximum number of timesteps $T$ is reached; or $\mathcal{C}$) when the policy hits the available budget for simulator calls $L$. To incentivize reducing the number of simulator calls, rewards $r(s_t, a_t, s_{t+1})$ are $0$ if $a_t = 0$ and $-1$ if $a_t = 1$. Additionally, a reward penalty is added when $\mathcal{B}$) or $\mathcal{C}$) occur to promote termination. The penalty is $-(L - l_t) - 1$ when $\mathcal{B}$) occurs and $-1$ when $\mathcal{C}$) occurs. This ensures the sum-of-rewards for non-terminating episodes is $-L - 1$. We have observed that using reward penalties based on $l_t$ rather than $t$ improves training stability. We refer the reader to Appendix D.3 for further details concerning the reward design.

**Local Surrogate**   The decision to perform a simulator call should rely on the quality of the local surrogate. A surrogate that is well-fitted to the simulator at the current $\boldsymbol{\psi}$ will presumably provide useful gradients, so gathering additional data and retraining is unnecessary. Vice-versa, a badly fitted surrogate will likely not provide useful gradients and may be worth retraining, even if a simulator call is expensive. We use the uncertainty feature $\sigma$ to provide this information.

**Local Surrogate Ensemble**   To construct $\sigma$, we replace the local surrogate with an ensemble of local surrogates, all trained on and applied to the same input data. The use of an ensemble empowers our approach with the ability to estimate uncertainties while avoiding the need to train a Bayesian posterior network (Fort et al., 2019; Lakshminarayanan et al., 2017; Wilson and Izmailov, 2020).

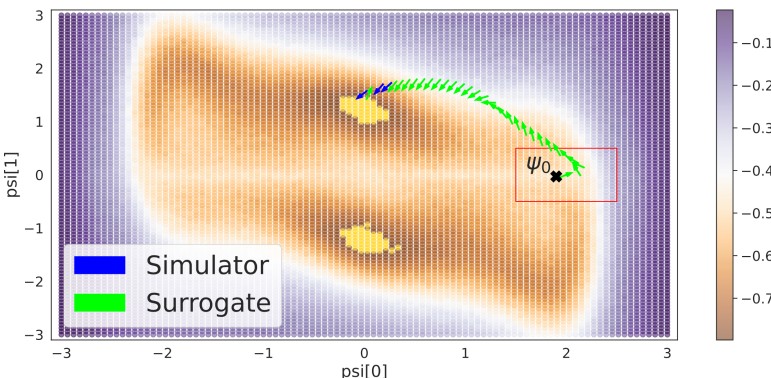

**Figure 2: Loss landscape and learned optimization trajectory for the Probabilistic Three Hump problem**. The yellow region denotes $\psi$ values that lead to termination. The $\epsilon = 0.5$ neighbour around $\psi_0$ (black cross) is visualized as the red box. Light green and blue arrows represent gradients from the surrogate or after a simulator call, respectively.

Each surrogate is implemented as a two-layer Multi-Layer Perceptron (MLP) with Rectified Linear Unit (ReLU) activation function. With such small models, the additional resource requirement for training an ensemble instead of a single surrogate is negligible. As the input, we use the tuple $(\psi, x, z)$, where $z$ is sampled from diagonal Normal distribution.

**Uncertainty Feature** We compute the prediction mean per surrogate on $D$ samples as $\bar{y} = \frac{1}{D} \sum_{i=1}^{D} [f_\phi(\psi, x_i, z_i)]$, and construct $\sigma$ as the standard deviation over these mean predictions. Specifically, $z$ accounts for the stochasticity of $f_{\text{sim}}$. Such an idea allows us to dramatically simplify the surrogate architecture compared to Shirobokov et al. (2020). Training GANs (Shirobokov et al., 2020) is notoriously more challenging than training a shallow MLP due to instabilities and mode collapse. Nonetheless, our "simpler" surrogate has enough capacity to locally approximate highly complex stochastic, and non-stochastic, simulators. Gradient steps in $\psi$ for simulator optimization are taken by using the average gradient estimated from the ensemble. See Appendix B.2 for further details concerning models implementation.

## 5 EXPERIMENTAL RESULTS

To assess the performance of our method, we test it on two different types of experiments. First, we consider stochastic versions of benchmark functions available in the optimization literature (Jamil and Yang, 2013). We consider the Probabilistic Three Hump, the Rosenbrock, and the Nonlinear Submanifold Hump problems. These benchmark functions are relevant for two reasons: (i) they allow us to compare our models against baselines on similar settings as in (Shirobokov et al., 2020); and (ii) they allow to easily gain insights into models performance. Since the Probabilistic Three Hump problem is two dimensional, i.e. $\psi \in \mathbb{R}^2$, we are able to especially conveniently visualize the objective landscape as well as the optimization trajectories. Furthermore, the Rosenbrock and Nonlinear Submanifold Hump problems allow us to test our approach on high-dimensional, more complex, problems before moving to real-world black-box simulators. The second type of experiments concerns real-world black-box simulators. We consider applications from two different scientific fields, namely the *Indoor Antenna Placement* problem for wireless communications and the *Muon Background Reduction* problem for high energy physics.

**Baselines** We compare our method against three baselines. We consider Bayesian optimization using Gaussian processes with cylindrical kernels (Oh et al., 2018), numerical differentiation with gradient descent, and local surrogate-based methods (L-GSO) (Shirobokov et al., 2020). Furthermore,

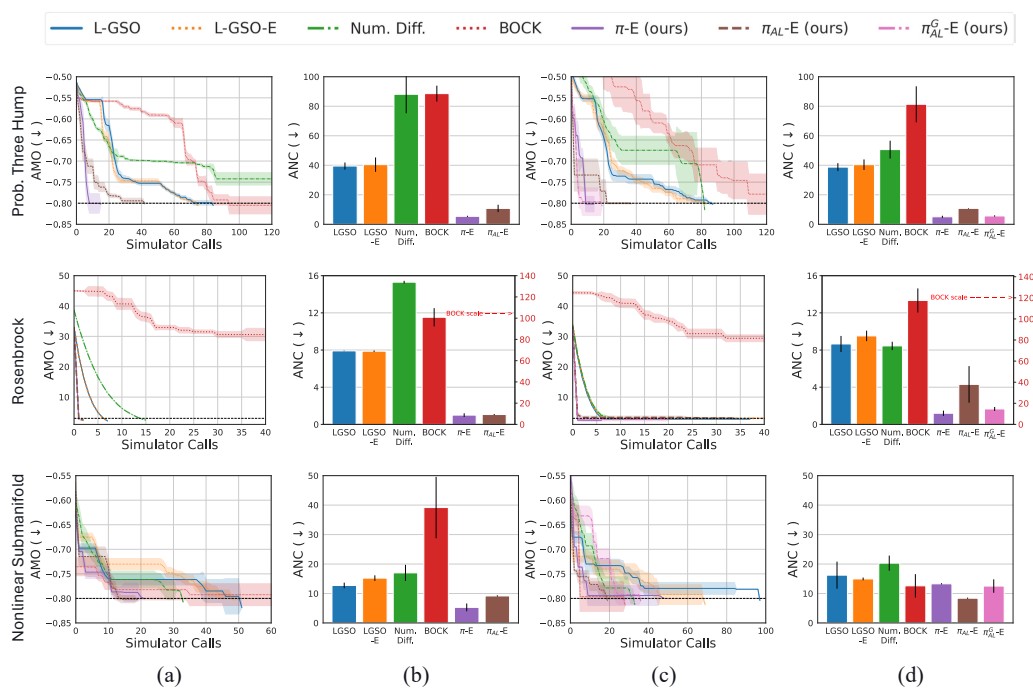

**Figure 3: Benchmark function results**. **Top** row: Probabilistic Three Hump problem. **Middle** row: Rosenbrock problem. **Bottom** row: Nonlinear Submanifold Hump Problem. AMO (the lower the better) on (a) a fixed and (c) a parameterized $x$ distribution. ANC (the lower the better) on (b) a fixed and (d) a parameterized $x$ distribution.

to guarantee a fair comparison against our models, we formulated an ensemble version[4] (L-GSO-E) of the local surrogate for L-GSO.

**Our Models** Policy methods are split into those that only output *when* to perform a simulator call, $\pi$-E, and those that also output *how* to sample $\psi$ values (for surrogate training) by providing the neighbour size $\epsilon$ that parameterizes the acquisition function $\pi_{AL}$-E. L-GSO, its ensemble version (L-GSO-E), and $\pi$-E use a fixed value for $\epsilon$ that depends on the problem at hand (see Appendix D.1). Finally, $\pi_{AL}^G$-E is a version of $\pi_{AL}$-E where the surrogate ensemble is always warm-started from the previous training step, such that the surrogate is continuously improved along the observed trajectories through $\psi$-space (see Appendix B.1 for more details).

**Metrics** We report experimental results by using two different metrics: the Average Minimum of the Objective function (AMO) for a specific budget of simulator calls and the Average Number of simulator Calls (ANC) required to terminate an episode. The first quantity answers the question: *What is the lowest value for the objective function achievable for a given budget of simulator calls?*; that might be used as an indicator of the efficacy of each simulator call. The second quantity answers the question: *What is the simulator call budget required, on average, to solve a black-box optimization problem?*; that might indicate how good the policy is at leveraging the surrogate and understanding its reliability. Therefore, those two metrics allow us to benchmark our approach against others by looking at relevant quantities (see Appendix D.5 for more details). In all experiments, uncertainties are quantified over evaluation episodes and different random seeds.

---

[4]L-GSO-E averages the gradients over the ensemble the same way our method does. The model does not leverage any uncertainty since it always calls the black-box simulator.

## 5.1 BENCHMARK FUNCTIONS

We consider a fixed and a parameterized input distribution for each benchmark function. Specifically, the latter setup corresponds to solving an entire family of related optimization problems each characterized by a different input distribution, $q_i(\boldsymbol{x})$. During training and evaluation of the policy, each episode is characterized by a different input distribution. In what follows we report the definition for each benchmark function only. A more detailed description can be found in Appendix D.1.

**Probabilistic Three Hump Problem**    As mentioned in the introduction to the section, the Probabilistic Three Hump problem concerns the optimization of a 2-dimensional vector $\boldsymbol{\psi}$. Specifically, the goal is to find $\boldsymbol{\psi}^*$ such that: $\boldsymbol{\psi}^* = \arg\min_{\boldsymbol{\psi}} \mathbb{E}[\mathcal{L}(y)] = \arg\min_{\boldsymbol{\psi}} \mathbb{E}[\sigma(y - 10) - \sigma(y)]$, where $\sigma$ is the sigmoid function and $y$, the observations vector, is given by: $y \sim \mathcal{N}(y;\ \mu_i, 1),\ i \in \{1, 2\}$. Being $\boldsymbol{\psi}$ a 2-dimensional vector, its optimization trajectory is amenable to visualization. Figure 2 illustrates that a fully trained policy can exploit the local-surrogate as much as possible and only perform a simulator call when the model is far from the initial training location (red square in Figure 2). Intuitively, such behaviour is foreseen. The surrogate model is expected to provide meaningful gradients in proximity to the $\boldsymbol{\psi}$ region where it was previously trained. However, as we move away from that region, we expect the quality of the gradients to decline until a simulator call is triggered and the local-surrogate re-trained. However, moving away from the last training region is not the sole condition that might trigger a simulator call. For instance, towards the end of the trajectory, the policy decides to call the simulator twice to gather more data to train the surrogate and then calls the simulator again before ending the episode, indicating that a rapidly changing loss landscape may also trigger a simulator call.

**Rosenbrock Problem**    In the Rosenbrock problem, we aim to optimize $\boldsymbol{\psi} \in \mathbb{R}^{10}$ such that: $\boldsymbol{\psi}^* = \arg\min_{\boldsymbol{\psi}} \mathbb{E}[\mathcal{L}(y)] = \arg\min_{\boldsymbol{\psi}} \mathbb{E}[y]$; where $y$ is given by: $y \sim \mathcal{N}(y;\ \gamma + x,\ 1)$, where $\gamma = \sum_{i=1}^{n-1} \left[ (\psi_i - \psi_{i+1})^2 + (1 - \psi_i)^2 \right]$.

**Nonlinear Submanifold Hump Problem**    This problem share a similar formulation to the Probabilistic Three Hump problem. However, the optimization is realized by considering the embedding $\hat{\boldsymbol{\psi}} = \boldsymbol{B} \tanh(\boldsymbol{A}\boldsymbol{\psi})$, where $\boldsymbol{A} \in \mathbb{R}^{16 \times 40}$ and $\boldsymbol{B} \in \mathbb{R}^{2 \times 16}$, of the vector $\boldsymbol{\psi} \in \mathbb{R}^{40}$. Subsequently, $\hat{\boldsymbol{\psi}}$ is used in place of $\boldsymbol{\psi}$ in the Probabilistic Three Hump problem definition.

## 5.2 REAL-WORLD SIMULATORS

We now focus on real-world optimization problems involving computationally expensive, non-differentiable black-box simulators. First, we look at the field of wireless communications considering two settings with a (non-stochastic) wireless ray tracer (Inc., 2023). Then, we move to the world of subatomic particles and solve a detector optimization problem for which we use the high energy physics toolkits Geant4 (Agostinelli et al., 2003) (stochastic simulator) and FairRoot (Al-Turany et al., 2012).

**Wireless Communication: Indoor Transmitting Antenna Placement**    We study the problem of optimally placing a transmitting antenna in indoor environments to maximize the signal strength at multiple receiver locations. Determining the signal strength in such a scenario typically requires a wireless ray tracer (Inc. (2023) in our case), which takes as input the transmit location candidate $\boldsymbol{\psi} \in \mathbb{R}^3$, alongside other parameters (e.g., receive locations, 3D mesh of scene). To predict the signal strength for a particular *link* (i.e., a transmit-receive antenna pair), the ray tracer exhaustively identifies multiple propagation paths between the two antennas and calculates various attributes of each path (e.g., complex gains, time-of-flight). The signal strength is computed from the coherent sum of the complex-valued gains of each path impinging on the receive antenna and is represented in log-scale (specifically, dBm). Optimally placing the transmit antenna is typically slow, as it amounts to naively and slowly sweeping over transmit location choices $\boldsymbol{\psi}$ and observing the simulated signal strengths. Instead, we employ our approach to "backpropagate" through the the surrogate and perform gradient descent steps on the location $\boldsymbol{\psi}$. Specifically, we consider two indoor scenes for this experiment and investigate how to use our approach to find an optimal transmit location that maximizes signal strength in the 3d scene (column (a) in Figure 4). The end goal in both cases is to find an optimal

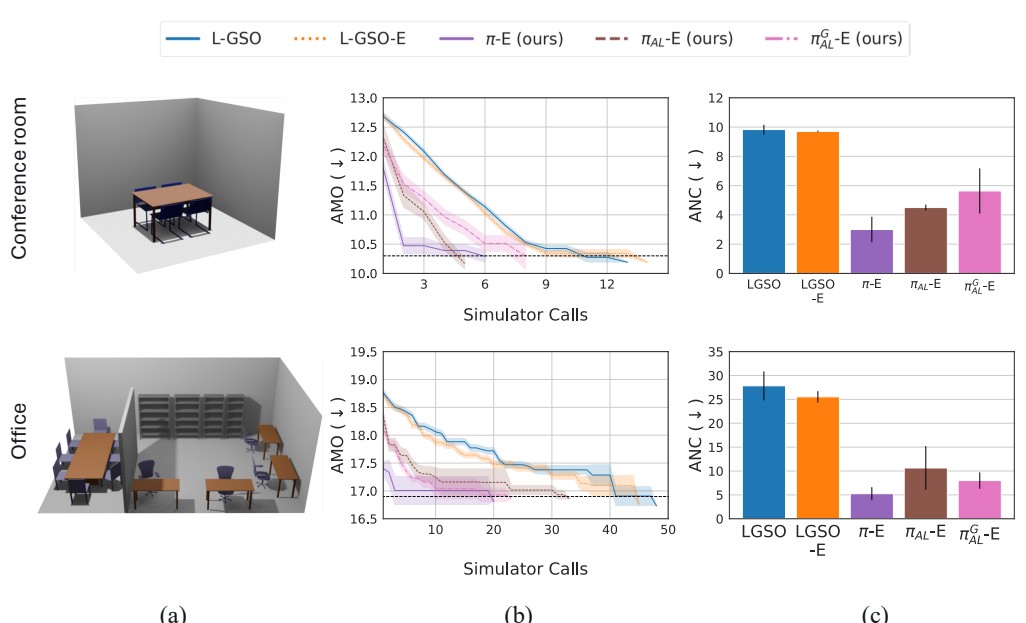

(a)               (b)               (c)

**Figure 4: Wireless ray-tracing results**. (a) Rendering of the indoor environment, (b) AMO (the lower the better), and (c) ANC (the lower the better). **Top** row: Conference room environment. **Bottom** row: Office room environment.

transmit antenna location $\psi$ that maximizes the median signal strength calculated over a distribution of receive locations $\boldsymbol{x} \sim q(\boldsymbol{x})$ (see Appendix D.2 for more details concerning simulations).

**Physics: Muon Background Reduction**  We consider the optimization of the active muon shield for the SHiP experiment (Baranov et al., 2017b). Typically, optimizing a detector is a crucial step in designing an experiment for particle physics. For instance, the geometrical shape, the intensity and orientation of magnetic fields, and the materials used to build the detector play a crucial role in defining the detector's "sensitivity" to specific types of particle interactions, i.e. events. Observed events are usually divided into signal, i.e., interactions physicists are interested in studying, and "background", i.e., events that are not of any interest and that might reduce the detector's sensitivity. Concerning the SHiP experiment, muons represent a significant source of background; therefore, it is necessary to shield the detector against those particles. The shield comprises six magnets, left image in Figure 5, each described by seven parameters. Hence, $\psi \in \mathbb{R}^{42}$. To run the simulations, we use the Geant4 (Agostinelli et al., 2003) and FairRoot (Al-Turany et al., 2012) toolkits. The input distribution $\boldsymbol{x}$ describes the properties of incoming muons[5]. Specifically, as in (Shirobokov et al., 2020), we consider the momentum ($P$), the azimuthal ($\phi$) and polar ($\theta$) angles with respect to the incoming $z$-axis, the charge $Q$, and $(x, y, z)$ coordinates. The goal is to minimize the expected value of the following objective function: $\mathcal{L}(\boldsymbol{y}; \boldsymbol{\alpha}) = \sum_{i=1}^{N} \mathbb{I}_{Q_i=1} \sqrt{(\alpha_1 - (\boldsymbol{y}_i + \alpha_2))/\alpha_1} + \mathbb{I}_{Q_i=-1} \sqrt{(\alpha_1 + (\boldsymbol{y}_i - \alpha_2))/\alpha_1}$

where $\mathbb{I}$ is the indicator function, $\alpha_1$ and $\alpha_2$ are known parameters defining the sensitive region of the detector, and $Q$ and $\boldsymbol{y}$ represent the electric charge and the coordinates of the observed muons, respectively. Minimizing $\mathcal{L}(\boldsymbol{y}; \boldsymbol{\alpha})$ corresponds to minimize the number of muons hitting the sensitive region of the detector.

## 5.3 Results & Discussion

On the problems involving benchmark simulators, policy-based methods achieve the best overall performance in both the fixed and parameterized $\boldsymbol{x}$-distribution scenarios, as shown in Figure 3. The $\pi$-E model scores best especially on average number of simulator calls (ANC) required to terminate an episode (bar plots in the figures) for almost all settings. Notably, using our trained policies, we

---

[5]Concerning the muon distribution, we use the same dataset as in Shirobokov et al. (2020). The dataset is available for research purposes.

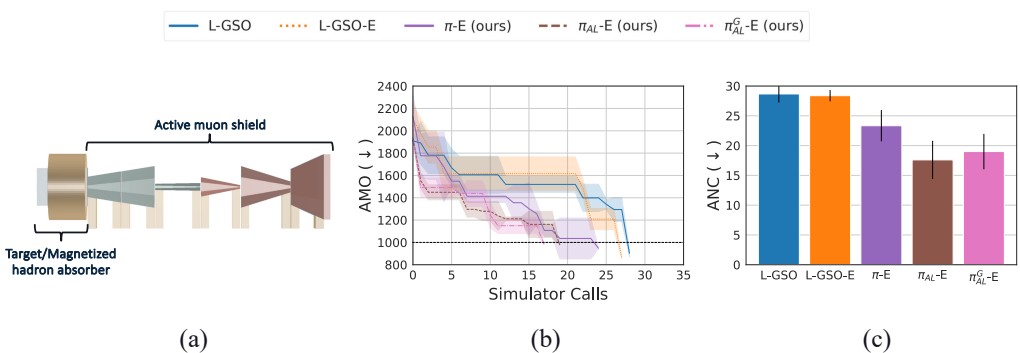

(a)  (b)  (c)

**Figure 5: Physics experiments results**. (a) Schematic view[6]of the active muon shield baseline configuration. The "Target/Magnet hadron absorber" details are not relevant to the current discussion and are reported for completeness only. See (Baranov et al., 2017a) for more details. (b) AMO (the lower the better), (c) and ANC (the lower the better).

observe a significant reduction in the number of required *simulator calls* of up to $\sim 90\%$ with respect to L-GSO. While the $\pi_{AL}$-E and $\pi_{AL}^G$-E models outperform all the baselines as well, there is no clear advantage compared to $\pi$-E. Note that the trust region size $\epsilon$ in $\pi$-E and L-GSO is set to the optimal value reported in Shirobokov et al. (2020) for these benchmark functions, which simplifies the problem relative to $\pi_{AL}$-E and $\pi_{AL}^G$-E. We note however that the warm-started surrogate of $\pi_{AL}^G$-E improves over $\pi_{AL}$-E in those cases, potentially by mitigating this difficulty. Similarly, the AMO evaluations show that our policies outperform previous methods, with the single exception of the Nonlinear Submanifold Hump problem; here our models are within error ranges of the best observed AMO. As noted in (Shirobokov et al., 2020), the BOCK baseline struggles in solving the Rosenbrock problem (Figure 3, middle row), likely due to the high curvature of the objective function under analysis. On the other hand, numerical differentiation appears to be less affected by this issue, thus reporting acceptable results for all the problems involving benchmark functions.

Given that the local surrogate baseline (L-GSO) generally outperforms the other baselines (and is on-par in the worst case), we use both its variants as our baseline method for experiments involving real-world black-box simulators. In these experiments, we see again that policy-based methods achieve the best performance in terms of both AMO and ANC. However, in contrast to the results of Figure 3, here the three different policy methods are very close to performing within error ranges of each other (Figure 4). In the particle physics experiment, $\pi_{AL}$-E and $\pi_{AL}^G$-E perform better on average than $\pi$-E (Figure 5). This could suggest that the true advantage of learning to adapt the trust region size, as in done by $\pi_{AL}$-E and $\pi_{AL}^G$-E, is only revealed in more complex optimization problems, such as the optimization of a detector for high energy physics experiments. We leave further investigation into assessing the potential advantages of learning the sampling strategy to future work. We refer to Appendix A for a more comprehensive discussion of limitations and future work.

## 6 CONCLUSION

We propose a novel method for minimizing the number of *simulator calls* required to solve optimization problems involving black-box simulators using (local) surrogates. The core idea of our approach is to learn an active learning policy that controls *when* the black-box simulator is used and *how* to sample data to train the local surrogates. We describe three policy model variations and present experiments showing they outperform previous methods, including local surrogate methods (Shirobokov et al., 2020), numerical differentiation, and Bayesian approaches (Oh et al., 2018) in a variety of setups that include benchmark functions and real-world black-box simulators. In particular, we observe a significant reduction in the number of *simulator calls* of up to $\sim 90\%$. Our results suggest that local surrogate-based optimization of problems involving black-box forward processes benefits from the guidance of both simple policies and learned sampling strategies.

---

[6]Image from (Baranov et al., 2017a). IOP Publishing, 2017, by Baranov, A., et al. Licensed under CC BY 3.0

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

## A    BROADER IMPACT, LIMITATIONS AND FUTURE WORKDS

**Broader Impact**    This paper proposes a novel policy-based approach to guide local surrogate-based problem optimization with black-box simulators. We believe the potential societal consequences of our work are chiefly positive, as it has the potential to promote the use of policy-based approaches in various scientific domains, particularly concerning optimization procedures involving black-box, non-differentiable, forward processes. However, it is crucial to exercise caution and thoroughly comprehend the behaviour of the models to obtain tangible benefits.

**Limitations and Future Works**    Gradient-based optimization may get stuck in local optima of the loss surface $\mathbb{E}_{p(\boldsymbol{y}|\boldsymbol{\psi},\boldsymbol{x})}\left[\mathcal{L}(\boldsymbol{y})\right]$. Investigating whether introducing a policy into the optimization can help avoid such local minima is an interesting direction of future research. The Probabilistic Three Hump problem has no local minima but does contain a few flat regions, where gradient-based optimization is more challenging. Exploratory experiments have provided weak evidence that the policy may learn to avoid such regions.

Hyperparameter tuning has mostly involved reducing training variance through tuning the number of episodes used for a PPO iteration, as well as setting learning rates and the KL-threshold. Little effort has been spent optimizing the policy or surrogate architectures; we expect doing so to further improve performance. Similarly, while PPO with a value function critic is a widely used algorithm, more recent algorithms may offer additional advantages, such as improved planning and off-policy learning for more data-efficient training (Haarnoja et al., 2018; Neumann et al., 2023).

## B    IMPLEMENTATION DETAILS

### B.1    POLICY

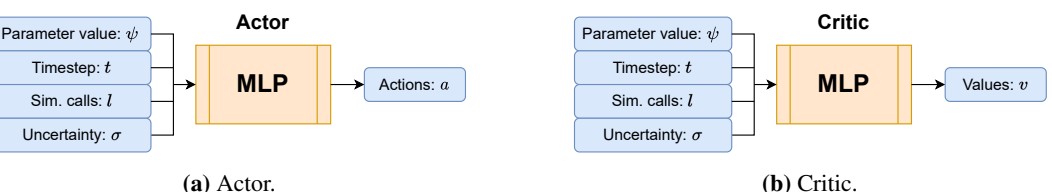

**(a)** Actor.                                    **(b)** Critic.

**Figure 6:** Schematic for the policy architecture. The policy consists of a separate Actor and Critic, which are both MLPs. They take $(\boldsymbol{\psi}, t, l, \sigma)$ as input and output the actions and value function estimates. Actions always contain the decision $b$ to perform a simulator call or not and may optionally also contain a value $\epsilon$ used for surrogate training data sampling.

The policy $\pi_\theta$ is composed of two separate neural networks: an Actor and a Critic. Both networks are ReLU MLPs with a single hidden layer of 256 neurons, schematically depicted in Figure 6. The input to both networks is the tuple: $(\boldsymbol{\psi}_t, t, l_t, \sigma_t)$, where $\boldsymbol{\psi}_t$ is the current parameter value (at timestep $t$), $l_t$ is the number of simulator calls already performed this episode, and $\sigma_t$ is the standard deviation over the average surrogate predictions in the ensemble.

The Actor outputs either one or three values. The first value is passed through a sigmoid activation and treated as a Bernoulli random variable, from which we sample $b$, representing the decision to perform a simulator call or not. If the policy outputs three values, the second and third values are treated as the mean and standard deviation of a lognormal distribution from which we sample $\epsilon$, the trust region size, for the current timestep. The standard deviation value is passed through a softplus activation function to ensure it is positive.

The Critic outputs a value-function estimate $V_\theta(s)$, where $\theta$ are policy parameters. We use this estimate to compute advantage estimates in PPO, as explained in detail in section C.1. Since rewards have unity order of magnitude, we expect return values to be anywhere in $[-T, 0]$. To prevent scaling issues, we multiply the Critic output values by $T$ before using them for advantage estimation.

**The $\pi_{AL}^G$ method**    When training a policy for downstream optimization of many related black-box optimization problems, it may be helpful to train a global surrogate simultaneously for such a problem

setting. Such a global surrogate might provide better gradients for problem optimization, especially if it has been jointly optimized with the policy. We have implemented the $\pi_{AL}^G$ method to test this. Here, the policy outputs both the decision to perform a simulator call and the trust-region size, $\epsilon$, just as in $\pi_{AL}$. However, the surrogate ensemble is "warm-started" from the previous training step every time a retraining decision is made. This results in a continuously optimized surrogate ensemble for the training trajectories. To prevent the surrogate from forgetting old experiences too quickly, we employ a replay buffer that undersamples data from earlier iterations geometrically. Specifically, when training the surrogate with trust-region $U_\epsilon^\psi$, we include all data inside $U_\epsilon^\psi$ for the current episode, half of the data inside $U_\epsilon^\psi$ from the previous episode, a quarter of the data seen two episodes ago, and so on.

## B.2 SURROGATE

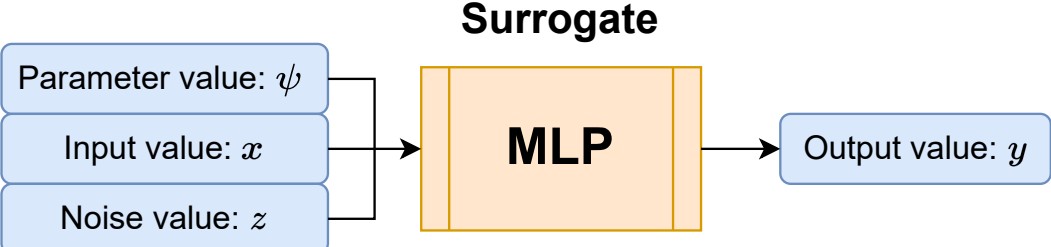

**Figure 7:** Schematic for the surrogate architecture. The surrogate is an MLP trained to mimic the simulator. It takes $(\psi, x, z)$ as input and outputs $y$.

Each surrogate model consists of a ReLU MLP with two hidden layers of 256 neurons that takes as input $(\psi, x, z)$ and outputs $y$. $z$ is sampled from a 100-dimensional diagonal unit Normal distribution. The surrogate architecture is schematically depicted in Figure 7.

Surrogates are trained on data generated from $f_{\text{sim}}$. Following the approach outlined in Shirobokov et al. (2020), we sample $M$ values $\psi_j$ inside the box $U_\epsilon^\psi$ around the current parameter value using an adapted Latin Hypercube sampling algorithm. For each of those $\psi_j$, we then sample $N = 3 \cdot 10^3$ $x$-values. We use $M = 5$ for the Probabilistic Three Hump problem, $M = 16$ for the Rosenbrock problem, and $M = 40$ for the Nonlinear Submanifold Hump problem. As in Shirobokov et al. (2020), this means a single "simulator call" consists of $1.5 \cdot 10^4$ function evaluations for Probabilistic Three Hump, $4.8 \cdot 10^4$ for Rosenbrock, and $6.0 \cdot 10^4$ for the Nonlinear Submanifold Hump.

To train the surrogates, we use the Adam optimizer for two epochs with a learning rate of $10^{-3}$ and a batch size of 512. Each *surrogate ensemble* comprises three surrogates, each trained on identical data but a different random seed. The uncertainty feature $\sigma$ is computed using mean predictions of each individual surrogate in the ensemble. Specifically, we compute the prediction mean per surrogate on $D$ samples as $\bar{y} = \frac{1}{D} \sum_{i=1}^{D} [f_\phi(\psi, x_i, z_i)]$, and construct $\sigma$ as the standard deviation over these mean predictions. We use $D = 100$ in all our experiments.

## B.3 FULL OBSERVABILITY OF THE MDP

Because the state of our reinforcement learning framework consists of the fully observed variables $(\psi_t, t, l_t, \sigma_t)$, we have formulated it as an MDP rather than as a POMDP (Partially Observable MDP). Concretely, our method can be applied if the parameter setting $\psi$ of a simulator is known at all times, since $t$ and $l_t$ increment based on policy decisions and $\sigma_t$ is generated using separate surrogate models. Training these surrogate models requires $(\psi, x, z)$, where $x$ and $x$ are user-generated and $y$ is observed simulator output. If $\psi$ is not observed, then backpropagation through the surrogate w.r.t. $\psi$ is not possible, and our method is not applicable. However, note that – even in black-box optimisation settings – the simulator parameter settings $\psi$ are generally input values specified by the user, and thus observed.

## C  TRAINING DETAILS

### C.1  TRAINING

We train our policy in an episodic manner by accumulating sequential optimization episodes and updating the policy using Proximal Policy Optimization (PPO) (Schulman et al., 2017) with Generalised Advantage Estimation (GAE) advantages (Schulman et al., 2016) (discount factor $\gamma = 1.0$, GAE $\lambda = 0.95$). Episodes terminate once any of the following conditions is met: A) the target value for the loss, $\tau$, has been reached, B) the number of timesteps $T = 1000$ has been reached, or C) the number of simulation calls $L = 50$ has been reached. For every training iteration, before doing PPO updates, we accumulate: 10 episodes for the Nonlinear Sub. Hump and 16 episodes for the Rosenbrock and Prob. Three Hump problems, 10 episodes concerning the wireless simulations and 5 for the high energy physics experiments. The different choices in the number of episodes to accumulate are mainly dictated by the time required to complete one episode.

We use the PPO-clip objective (with clip value $0.2$) on full trajectories with no entropy regularization to perform Actor updates. We perform multiple Actor updates with the same experience until either the empirical KL-divergence between the old and new policy reaches a threshold ($3 \cdot 10^{-3}$ for simulator-call decision actions, $10^{-2}$ for trust-region size $\epsilon$ actions), or 20 updates have been performed. In practice, we rarely perform the full 20 updates. Updates use the Adam optimizer with learning rate $3 \cdot 10^{-4}$.

Similarly, we perform multiple Critic updates using the Mean-Squared Error (MSE) between the estimated values $V_\theta(s_t)$ and the observed return (sum of rewards, as $\gamma = 1.0$) $R_t$ at every timestep. We keep updating until either MSE $\leq 30.0$ or ten updates have been done. This approach helps the critic learn quickly initially and after seeing surprising episodes but prevents it from over-updating on similar experiences (as MSE will be low for those iterations). Updates use Adam with learning rate $10^{-4}$. See Algorithm 1 for the training pseudo-code and Algorithm 2 for the evaluation procedure pseudo-code.

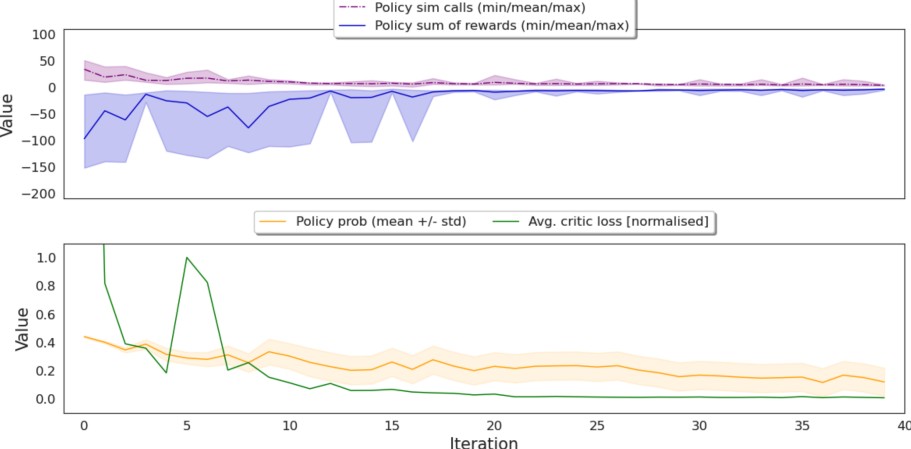

**Figure 8: Top**: Average number of simulator calls (purple curve) and average sum of rewards (blue curve) after each PPO iteration. Values are averaged across evaluation episodes. The upper and lower bound for the shadowed areas represent the max. and min. for each of the two mentioned metrics, respectively. **Bottom**: Average value of the probability of calling the black-box simulator (orange curve) and average critic loss (green curve). Values are averages across evaluation episodes for each PPO iteration.

To assess the performance of our models, we run 32 evaluation episodes for the benchmark functions and 20 and 5 evaluation episodes for the wireless and physics experiments, respectively. Moreover, we consider three random seeds for L-GSO and policy models, while we used ten random seeds for the BOCK and Num. Diff. baselines.

---

**Algorithm 1:** Training the (active learning) policy.

**Data:** Simulator $f_{\text{sim}}(\boldsymbol{\psi}, \boldsymbol{x})$; surrogate $f_\phi(\boldsymbol{\psi}, \boldsymbol{x})$; objective function $\mathcal{L}$; policy $\pi_\theta$; number $N$ of $\boldsymbol{\psi}$ to sample when training the surrogate; number $M$ of of $\boldsymbol{x}$ to sample for each $\boldsymbol{\psi}$; distributions $Q(q)$ over distributions $q(\boldsymbol{x})$ to sample $\boldsymbol{x}$ from; initial value $\boldsymbol{\psi}_0$; target function value $\tau$; number $T$ of timesteps to run each simulation for (episode-length); maximum number of simulator calls $L$; $\boldsymbol{\psi}$ optimiser OPTIM$_\psi$ with learning rate $\lambda$; number of policy training iterations $K$; number of episodes to accumulate for a PPO step $G$; policy optimiser OPTIM$_\pi$; reward function $\mathcal{R}$; experience buffer $B$; discount factor $\gamma$.

**for** $k \in (1, ..., K)$ **do**
  Empty experience buffer $B$.
  **for** $\_ \in (1, ..., G)$ **do**
    Initialise number of simulator calls done: $l_t \leftarrow 0$.
    Set return: $R \leftarrow 0$.
    Sample $\boldsymbol{x}$-distribution $q \sim Q$.
    **for** $t \in (1, ..., T)$ **do**
      Sample $\boldsymbol{x} \sim q(x)$.
      Obtain uncertainty $\sigma_t$ from the ensemble surrogate $f_\phi(\boldsymbol{\psi}_t, \boldsymbol{x})$.
      Construct state: $s \leftarrow (\boldsymbol{\psi_t}, t, l_t, \sigma_t)$.
      Obtain action: $a = (\texttt{do\_retrain}, \texttt{trust\_region\_size}) \leftarrow \pi_\theta(s)$.
      **if** `do_retrain` **then**
        Obtain $N$ samples $\boldsymbol{\psi}_n$ from trust region with size `trust_region_size`.
        Obtain $M$ samples $\boldsymbol{x}_m \sim q(\boldsymbol{x})$ for each of these $\boldsymbol{\psi}_n$.
        Combine into dataset $\{\boldsymbol{\psi}, \{\boldsymbol{x}\}^M\}^N$ and optionally filter or include data from previous timesteps.
        Retrain surrogate: $f_\phi$ on this dataset.
        Increment number of simulator calls: $l_t \leftarrow l_t + 1$.
      **end**
      Obtain surrogate gradients: $\boldsymbol{g}_t \leftarrow \nabla_\psi \mathcal{L}(f_\phi(\boldsymbol{\psi}, \boldsymbol{x}))|_{\boldsymbol{\psi}_t}$.
      Do optimisation step: $\boldsymbol{\psi}_{t+1} \leftarrow \text{OPTIM}_\psi(\boldsymbol{\psi}_t, \boldsymbol{g}_t, \lambda)$.
      `terminated` $\leftarrow \mathbb{E}\left[\mathcal{L}(f_{\text{sim}}(\boldsymbol{\psi}_t, \boldsymbol{x}))\right] \leq \tau$
      Obtain reward: $r \leftarrow \mathcal{R}(s, a, \boldsymbol{\psi}_{t+1})$.
      Store $(s, a, r)$ and any other relevant information in buffer $B$.
      **if** `terminated` **then**
       | break
      **end**
      **if** $l$ *equals* $L$ **then**
       | break
      **end**
    **end**
  **end**
  Update policy $\pi_\theta \leftarrow \text{OPTIM}(\pi_\theta, B, \gamma)$.
**end**

---

Figure 8 shows that the policy is actually able to learn *when* to call the simulator. Initially, during the first stages of the training, the policy generates completely random actions, resulting in an average probability of calling the simulator close to 0.5 (bottom plot in Figure 8). However, as the training progresses, such a probability gradually decreases, leading to a reduction in the number of simulator calls (top plot in Figure 8).

## C.2 OBJECTIVE LANDSCAPE AND OPTIMIZATION TRAJECTORY

Experiments with low-dimensional functions, such as the Probabilistic Three Hump problem ($\psi \in \mathbb{R}^2$), allow us to easily visualize optimization trajectories to gain insights into the models behaviour.

As mentioned in the main corpus of the paper, practitioners in many scientific fields may need to solve a set of related balck-box optimization problems that can become costly if each optimization process has to begin *ab initio*. Therefore, we investigated the robustness of the policy trained on a given setup, i.e. input $x$-distributions, and then tested on different ones. To mimic such a scenario, we consider a parameterized input $\boldsymbol{x}$-distribution. In real-world experiments, such a variation could correspond to different properties of the input data used to run the simulations. We already report the results concerning such tests in Section 5. In Figure 9, we show the optimization landscape for different $\boldsymbol{x}$-distributions for the Prob. Three Hump problem. It is worth noticing that, although the minima generally correspond to similar neighbours of the $\psi$ values, the landscape dramatically changes from one distribution to another.

## C.3 EXPERIMENTS COMPUTE RESOURCES

Performing a single optimization for the benchmark functions and the wireless experiments does not require a significant amount of computational resources and can be conducted using any commercially available NVIDIA GPU. A single optimization can be easily fitted on a single GPU. On the other hand, physics experiments require extensive computing resources for running simulations. While it is still feasible to run the entire optimization on a single machine, it might take a consistent amount of

---

**Algorithm 2:** Inference with the (active learning) policy.

---

**Data:** Simulator $f_s(\boldsymbol{\psi}, \boldsymbol{x})$; surrogate $f_\phi(\boldsymbol{\psi}, \boldsymbol{x})$; objective function $\mathcal{L}$; trained policy $\pi_\theta$; number
$N$ of $\boldsymbol{\psi}$ to sample when training the surrogate; number $M$ of of $\boldsymbol{x}$ to sample for each $\boldsymbol{\psi}$;
distributions $q(\boldsymbol{x})$ to sample $\boldsymbol{x}$ from; initial value $\boldsymbol{\psi}_0$; target objective value $\tau$; number $T$
of timesteps to run each simulation for (episode-length); maximum number of simulator
calls $L$; $\boldsymbol{\psi}$ optimizer $\text{OPTIM}_\psi$ with learning rate $\lambda$.

**for** $t \in (1, ..., T)$ **do**
    Initialise number of simulator calls done: $l_t \leftarrow 0$.
    Sample $\boldsymbol{x} \sim q(x)$.
    Obtain uncertainty $\sigma_t$ from the ensemble surrogate $f_\phi(\boldsymbol{\psi}_t, \boldsymbol{x})$.
    Construct state: $s \leftarrow (\boldsymbol{\psi_t}, t, l_t, \sigma_t)$.
    Obtain action: $a = (\texttt{do\_retrain}, \epsilon = \texttt{trust\_region\_size}) \leftarrow \pi_\theta(s)$.
    **if** *do\_retrain* **then**
        Obtain $N$ samples $\boldsymbol{\psi}_n$ from trust region with size $\epsilon$.
        Obtain $M$ samples $\boldsymbol{x}_m$ for each of these $\boldsymbol{\psi}_n$.
        Combine into dataset $\{\boldsymbol{\psi}, \{\boldsymbol{x}\}^M\}^N$ /* filter or include data from
            previous timesteps.                                 */
        Retrain surrogate: $f_\phi$ on this dataset.
        Increment number of simulator calls: $l_t \leftarrow l_t + 1$.
    **end**
    Obtain surrogate gradients: $\boldsymbol{g}_t \leftarrow \nabla_\psi \mathcal{L}(f_\phi(\boldsymbol{\psi}, \boldsymbol{x}))|_{\boldsymbol{\psi}_t}$.
    Do optimization step: $\boldsymbol{\psi}_{t+1} \leftarrow \text{OPTIM}_\psi(\boldsymbol{\psi}_t, \boldsymbol{g}_t, \lambda)$.
    $\texttt{terminated} \leftarrow \mathbb{E}\left[\mathcal{L}(f_s(\boldsymbol{\psi}_t, \boldsymbol{x}))\right] \leq \tau$
    **if** *terminated* **then**
        | break
    **end**
    **if** *l equals L* **then**
        | break
    **end**
**end**

---

time when simulating thousands of particles. The primary bottleneck for such experiments stems
from the Geant4 (Agostinelli et al., 2003) simulator, which is highly CPU-demanding. Since the
simulations of individual particles are independent of each other, they can be run in parallel without
communication between processes. In our experiments, we split up each simulation into chunks
of 2000 particles which resulted in run times of 5-15 minutes per simulation on single CPU core,
depending on the exact hardware.

## D   EXPERIMENTAL DETAILS

### D.1   BENCHMARK FUNCTIONS

Our tests with benchmark functions employ a probabilistic version of three benchmark functions
from the optimization literature: Probabilistic Three Hump, Rosenbrock, and Nonlinear Submanifold
Hump. The first one is a two-dimensional problem that lends itself well to visualization. Instead, the
$N$-dimensional Rosenbrock (with $N = 10$) and the Nonlinear Submanifold Hump problems are used
to test our method on higher-dimensional settings.

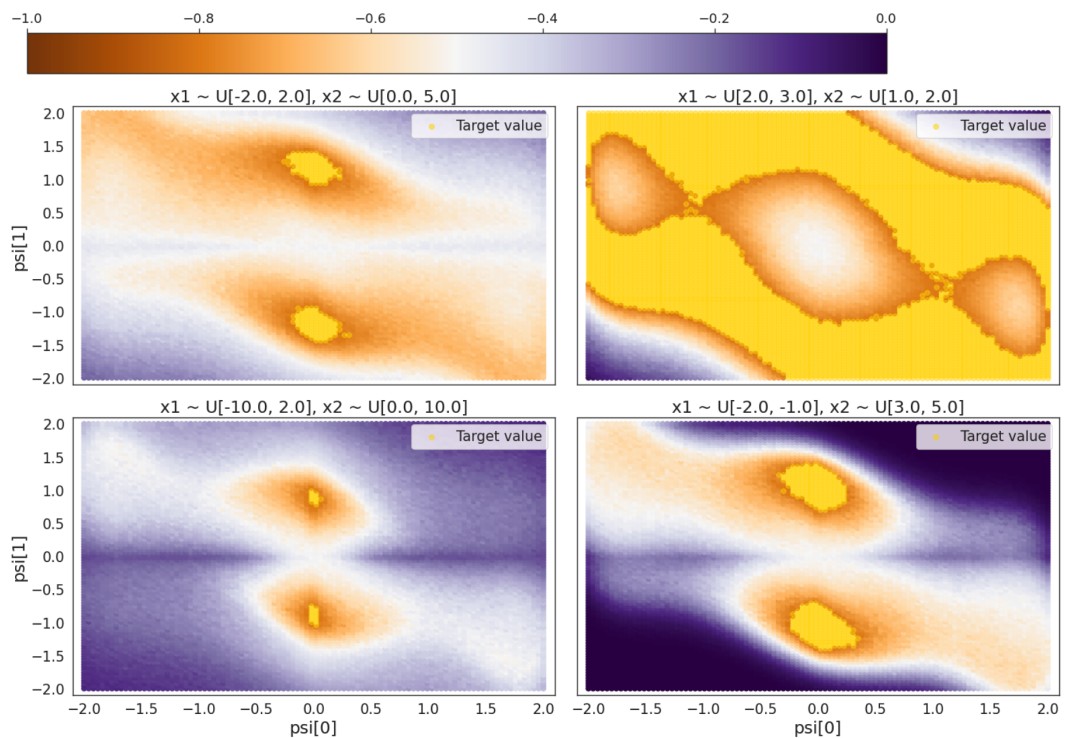

**Figure 9:** Loss landscape for the Probabilistic Three Hump Problem for different bounds on the $x$-distribution. Shown is $\frac{1}{n}\sum_{i=0}^{n-1}\mathcal{L}(f_s(\boldsymbol{\psi}, \boldsymbol{x}_i))$ for a grid of $\boldsymbol{\psi}$ values ($n = 100$). The yellow region denotes $\boldsymbol{\psi}$ values that lead to termination.

**Probabilistic Three Hump Problem**    The goal is to find the 2-dimensional $\boldsymbol{\psi}$ that optimizes:[7]

$$\boldsymbol{\psi}^* = \arg\min_{\boldsymbol{\psi}} \mathbb{E}\left[\mathcal{L}(y)\right] = \arg\min_{\boldsymbol{\psi}} \mathbb{E}\left[\sigma(y - 10) - \sigma(y)\right], \text{s.t.}$$

$$y \sim \mathcal{N}(y|\mu_i, 1),\ i \in \{1, 2\},\ \mu_i \sim \mathcal{N}(x_i h(\boldsymbol{\psi}), 1),\ x_1 \sim U[-2, 2],\ x_2 \sim U[0, 5], \tag{2}$$

$$P(i = 1) = \frac{\psi_1}{||\boldsymbol{\psi}||_2} = 1 - P(i = 2),\ h(\boldsymbol{\psi}) = 2\psi_1^2 - 1.05\psi_1^4 + \psi_1^6/6 + \psi_1\psi_2 + \psi_2^2.$$

We consider an episode terminated when $\mathbb{E}\left[\mathcal{L}(y)\right] = \frac{1}{N}\sum_{i=1}^N \mathcal{L}(f_{\text{sim}}(\boldsymbol{\psi}, \boldsymbol{x}_i)) \leq \tau = -0.8$, which we evaluate after every optimization step using $N = 10^4$ samples. Following (Shirobokov et al., 2020), we use $\epsilon = 0.5$ as the trust-region size. The optimization is initialized at $\boldsymbol{\psi}_0 = [2.0, 0.0]$; this is a symmetry point in the Three Hump function such that optimization with stochastic gradients can fall into either of the two wells around the two minima of the function. Such a procedure requires our methods to learn good paths to both optima, making the task more interesting. In principle, optimization could be initialized at any $\boldsymbol{\psi}_0$.

**Rosenbrock Problem**    The goal for this problem is to find the 10-dimensional $\boldsymbol{\psi}$ that optimizes:

$$\boldsymbol{\psi}^* = \arg\min_{\boldsymbol{\psi}} \mathbb{E}\left[\mathcal{L}(y)\right] = \arg\min_{\boldsymbol{\psi}} \mathbb{E}\left[y\right] \tag{3}$$

$$y \sim \mathcal{N}\left(y;\ \sum_{i=1}^{n-1}\left[(\psi_{i+1} - \psi_i^2)^2 + (\psi_i - 1)^2\right] + x,\ 1\right),\ x \sim \mathcal{N}(x; \mu, 1);\ \mu \sim U\left[-10, 10\right] \tag{4}$$

We consider an episode terminated when $\mathbb{E}\left[\mathcal{L}(y)\right] = \frac{1}{N}\sum_{i=1}^N \mathcal{L}(f_{\text{sim}}(\boldsymbol{\psi}, \boldsymbol{x}_i)) \leq \tau = 3.0$, which we evaluate after every optimization step using $N = 10^4$ samples. Following (Shirobokov et al., 2020), we use $\epsilon = 0.2$ as the trust-region size and $\boldsymbol{\psi}_0 = [\overrightarrow{2.0}] \in \mathbb{R}^{10}$ to initialize the optimization.

---

[7]Here the upper bound of $x_1$ and lower bound of $x_2$ are switched compared to the notation in Equation (3) of (Shirobokov et al., 2020). These bounds match the official implementation of L-GSO as of August 2023.

**Nonlinear Submanifold Hump Problem**   In this problem, we seek to find the optimal parameters vector $\psi$ in $\mathbb{R}^{40}$ by utilizing a non-linear submanifold embedding represented by $\hat{\psi} = B\tanh(A\psi)$, where $A \in \mathbb{R}^{16 \times 40}$ and $B \in \mathbb{R}^{2 \times 16}$. To achieve this, we use $\hat{\psi}$ in place of $\psi$ in the Probabilistic Three Hump problem definition. Also, for the current setup, we follow similar settings as in (Shirobokov et al., 2020): the orthogonal matrices $A$ and $B$ are generated via a $QR$-decomposition of a random matrix sampled from the normal distribution; we use $\epsilon = 0.5$ as the trust-region size and initialize the optimization at $\psi_0 = [2.0, \overrightarrow{0.0}] \in \mathbb{R}^{40}$.

**Parameterized Input Distribution**   In order to evaluate the generalization capabilities of our method, we further parameterize each target function by placing distributions on the bounds of the Uniform distributions from which $x_1$ and $x_2$ are sampled. We randomly sample new bounds in every episode to ensure that the policy is exposed to multiple related but distinct simulators during training and evaluation. Concerning the Hump problems, we sample the lower and upper bounds of $x_1$ from $\mathcal{N}(-2, 0.5)$ and $\mathcal{N}(2, 0.5)$, respectively. For $x_2$, we instead use $\mathcal{N}(0, 1)$ and $\mathcal{N}(5, 1)$. For the Rosenbrock problem, we sample the lower and upper bounds of $x$ from $\mathcal{N}(0, 2)$ and $\mathcal{N}(10, 2)$, respectively. Occasionally, an episode may not terminate as the specified termination value $\tau$ is below the minimum loss value for some samplings.

## D.2   Real-world Simulators

**Wireless Communication: Indoor Transmitting Antenna Placement**   The goal in this scenario is to find an optimal *transmit* antenna location $\psi$ that maximises the signal strength over multiple *receiving* antenna locations $x \sim q(x)$. Now, we detail aspects on the experimental setup for the experiments. We run wireless simulations using Matlab's Antenna Toolbox Inc. (2023), by evaluating the received signal strength (`sigstrength` function). The simulations are run in two 3d scenes ('`conferenceroom`' and '`office`'), both of which are available by default and we additionally let Matlab automatically determine the surface materials. We use the 'raytracing' propagation model with a maximum of two reflections and by disabling diffraction. The end-objective is to find a transmit antenna location $\psi$ maximize the received signal strength over locations $x \sim q(x)$. We constrain the locations in a 3d volume spanning the entire XY area of the two scenes: $3 \times 3$m in `conferenceroom` and $8 \times 5$m in `office`. The transmit elevations $\psi$ are constrained between 2.2-2.5m and 3.0-3.2m per scene, and the receive locations between 1.3-1.5m (identical for both scenes). The end-objective is to identify a transmit location $\psi$ such that the median receive signal strength is maximized over a uniform distribution of receive antenna locations $q(x)$.

## D.3   Reward Design

The reward function is chosen to incentivize the policy to reduce the number of simulator calls. This is achieved by giving a reward of -1 every time the policy opts to call the simulator, contrasting with a reward of 0 when it does not. However, with this reward function the policy could achieve maximum return (of zero) by never calling the simulator even if this leads to non-terminating episodes. An extra term is required to make any non-terminating episodes worse than any terminating one. Since the minimum return is $-L$, corresponding to the maximum number of simulator calls for an episode, that is achieved by setting a reward penalty of $-(L - l_t) - 1$ whenever the episode ends for reasons other than reaching the target value $\tau$: if the simulator call budget has been exhausted, then $l_t = L$ and the penalty is $-1$; if the timestep budget has been exhausted, then we have accumulated $-l_t$ return already. In both cases, adding this penalty leads to a total return of $-L - 1 < -L$.

## D.4   Termination Value

Termination values $\tau$ for the Probabilistic Three Hump and Rosenbrock problems are chosen to trade-off episode length and optimisation precision. Selecting values very close to the exact minimum value of the objective function $\mathcal{L}$ leads to extremely long episodes, due to the stochastic nature of the optimisation process. Moreover, parameterizing the distribution of the $x$ variables changes the (expected) objective value minimum, such that choosing a too low value for $\tau$ leads to episodes that cannot terminate even in theory. Computing the minimum of $\mathcal{L}$ on the fly for the various parameterizations of $x$ is not trivial, and so we opted for choosing a $\tau$ that generally suffices for good

performance across parameterizations of a given problem. These values are chosen by manually inspecting L-GSO runs.

### D.5 METRICS

As we mentioned in Section 5, we use two metrics to compare our models against the baselines: the Average Minimum of the Objective function (AMO) for a specific budget of simulator calls and the Average Number of simulator Calls (ANC) required to terminate an episode. We now delve deeper into both of them. The meaning of the latter is quite straightforward. We consider the average number of simulator calls to solve the problem. We compute the average across evaluation episodes and random seeds. In contrast, the AMO is slightly less intuitive to interpret. One might question whether the value of the ANC should align with the maximum value on the $x$-axes for the AMO. In other words, assuming that for a given model, the ANC is equal to, e.g. 10, *should one expect that at a value $x = 10$, the AMO will be equal to the termination value?* Generally speaking, the answer is *no*. To explain why that is the case, we can report the following example. Let us assume that, for a given model, we have the following three episodes, each characterized by a specific length and value of the objective function at each simulator call:

- Episode 1: [20, 12, 7, 5, 3, 1]
- Episode 2: [18, 6, 1]
- Episode 3: [15, 5, 1]

We assumed the target value, $\tau$, to be 1. For simplicity, we used integers for the objective value. As we can see from the example, we have ANC $= 4$. Now, if we examine the AMO for $x = 4$, we find that it is equal to 5 since only the first episode contributes to it, which is greater than $\tau$. Therefore, one cannot directly map the $x$-axis from the AMO to the $y$-axis of the ANC. Such a one-to-one mapping would exist only when all episodes always require the same number of simulator calls, which is not the case. We hope that our explanation has clarified the interpretation of the results we reported in the main corpus of the paper.

