# OpenReview forum: "Simulating, Fast and Slow: Learning Policies for Black-Box Optimization"
_ICLR.cc/2025/Conference — ICLR 2025 Conference Withdrawn Submission_

### Official Review · Reviewer_ipaY · 2024-10-24

**Soundness:** 2
**Presentation:** 3
**Contribution:** 2
**Rating:** 5
**Confidence:** 4

**Summary:**

This submission proposes a method to perform optimization of a loss function that takes the form of an expectation under a sampleable simulator, but whose (stochastic) gradient gradient is not analytically available.

To do so, the method follows an approach already explored in the field approximately minimizing the loss function using Stochastic Gradient Descent, where the gradient is obtained using a differentiable surrogate of the simulator. To produce a gradient estimate, at each gradient step, a policy network  determines whether to train a new simulator surrogate.
This policy network is trained in a separate algorithm prior to the optimization using a reward model which incentivises both reaching satisfying objective function values and minimizing the number of surrogate retraining operation, as the latter involves sampling from the simulator which may be costly in practice.

Variants of the method are constructed where the policy is also trained to estimate the width of the parameter neighborhood used to retrain the surrogate.

The method is showcased on a series of benchmarks and on real world experiments. In the case of the benchmark model, an additional "parameterized input distributions" setting is considered, where the goal is to be able to perform optimization of multiple loss functions, each of them with the same simulator function, but differing in which input distribution is used. The training of the policy network is amortized over all parameterized distributions.

**Strengths:**

- I am not an expert in the area, but the problem of amortizing an active learning strategy over multiple optimization is an interesting problem (although I am not sure how relevant this problem is in practice).
- The method is tested on multiple real-world experiments, and is able to handle stochastic and non-stochastic simulators, suggesting a potential high impact for practitioners.
- Aside from the issue related to amortization (discussed above), the presentation is very clear.

**Weaknesses:**

If I understood correctly, I see two ways in which the method could be beneficial:

1. either the total number of simulator calls over the 2 phases ((1) training the policy network and (2) optimizing the function) is smaller than the number of simulator calls in  other methods.
2. or, as training the policy network can be amortized over multiple input distributions q_i, if one needs to solve multiple such optimization problems using different input distributions, the cost of training ends up being negligible compared to the gains at inference (compared to other methods) that the training policy network provides.


However, both of these benefits are neither clearly described nor demonstrated in the main body:
- regarding point 1: when reporting the metrics, it is not mentioned whether training is accounted for
- regarding point 2: the use of multiple input distributions is mentioned, but once (l 326), and without stating out clearly why it is an important use case for the method at hand. If point 2 ends up being the main advantage of the method, I would expect the introduction to provide more motivation as of why parameterized input distributions matter in practice. Moreover, I could not find a mention of parameterized distributions in real-world experiments.


In my opinion, the paper needs to demonstrate clearly that the additional training step is justified. Due to the reasons described above, this demonstration remains limited, and thus I will recommend rejection until these issues are addressed. I may have missed additional benefits, in which case I will change my recommendation.

Some important training and evaluation details are missing from the main body, see questions.

**Questions:**

- how are the surrogate neural networks trained?
- what are the values employed for K and G?
- Do the plot reporting the number of simulator calls incorporates the cost of training? This is key for a fair comparison in my opinion.

Minor:

- Typo l. 403: "receive locations" -> "receiving locations"

---

### Official Review · Reviewer_Gry9 · 2024-11-01

**Soundness:** 1
**Presentation:** 2
**Contribution:** 1
**Rating:** 3
**Confidence:** 3

**Summary:**

The paper focuses on black-box optimization problems using a learnt RL policy. The policy is then used to guide the training of differentiable surrogate models, where the gradients of these surrogate models are used to perform black-box optimization. The paper compares to a local-surrogate based approach and a Bayesian optimization approach.

**Strengths:**

* The experiment relating to the antenna placement is interesting and could be the focus of an applications paper if expanded on.
* The motivation to work on black-box optimization and gradient-free approaches makes sense.
* The introduction is written well.
* It is good that the comparison is made with an ensemble of L-GSO models.

**Weaknesses:**

* The main weakness of the paper seems to be highlighted in the appendix, where algorithm 1 shows that the policy is required to be trained on simulated data prior to the actual black-box optimization in algorithm 2. However, this is not made clear in the main paper. This means that the results in the paper do not include the large number of simulations required to train a policy and seems to defeat the purpose of reducing the cost of simulation. Specifically, to train the policy, the simulator is queried in a triple nested for-loop, which implies the requirement of many queries before performing the black-box optimization.
* The section “Policy-Based Black-Box Optimization” does not highlight the technical approach of the paper. It continuously refers to the appendix instead of introducing the technical content. It should be the case that this is the most important section of the paper and therefore information regarding training and policy implementation (when it is key to the algorithm) needs to be made explicit in this section.
* The uncertainty feature is introduced and then not mentioned again. Is it necessary? Especially since the appendix mentions using an ensemble of 100 models. Therefore there is the potential for large overhead.
* It is not clear from the paper what the definition of a “parameterized input distribution” is, and why it might even be needed. Therefore, it is hard to interpret the difference between columns (a) and (c) in Figure 3.
* The results in Figure 3 do not match Shirobokov et al. (2020) which is a concern. For example at 100 function calls in Figure 2a of the prior paper, no algorithm has reached -0.8, but all algorithms have reached -0.8 in this paper except numerical differentiation. The difference can also be seen for the 10D Rosenbrock and the nonlinear submanifold hump problem. This needs to be addressed in the paper as to why there is this discrepancy.   There is also a similar discrepancy for the Physics experiment where L-GSO gets to an objective value of about 70 after 20 calls in Shirobokov et al. (2020).

Minor:
* Is “learning active learning” an accepted phrase in the literature? Maybe meta active learning as in Ravi and Larochelle (2018)?

**Questions:**

* What is the motivation for using an ensemble to build the $\sigma$? It does not seem to be explained whether it is effective compared to not using it.

---

### Official Review · Reviewer_n7DN · 2024-11-04

**Soundness:** 2
**Presentation:** 3
**Contribution:** 1
**Rating:** 5
**Confidence:** 4

**Summary:**

The authors introduce an strategy combining RL and neural network simulation surrogates to optimize simulation parameters. The basic idea is to train local surrogates to estimate simulator parameter gradients, while training a policy network that tells us whether to trust the surrogates or to retrain them on a new batch of simulations. Using this approach, they aim to accurately estimate simulation parameters while minimizing the number of calls to the simulator. They test this approach on a combination of benchmark and real-world tasks.

**Strengths:**

The approach is straightforward and intuitively appealing.

The presentation and structure and generally clear and informative (with a few exceptions noted below).

The ability to efficiently estimate simulation parameters is clearly important to many fields, and using local information is bound to be a winning strategy on some problems.

**Weaknesses:**

The fact that a single "simulator call" means calling $f_\text{sim}$ thousands of times should be prominently stated early on, and not hidden in the appendix. The figures give a false impression that the algorithms used here can work with dozens of $f_\text{sim}$ evaluations, while in fact they appear to require thousands, similar to SBI methods that do not rely on local gradient estimates. The total number of simulation calls required to train the policy network should also be reported. If we simply trained a network to map from observations to parameters using a least squares loss and this number of samples from the parameter prior, then evaluated the loss, would it actually be worse than the reported results for inference (while using zero simulator calls at inference)?

More generally because the simulation costs are comparable, this method should be directly compared to SBI methods relying on neural density estimations, for which there are mature implementations (Tejero-Cantero 2020) and benchmarks (Lueckmann 2021). Can the proposed method solve these problems, and how sample efficient is it compare to SNPE, SNLE etc.? Because the software is readily available, this should be straightforward to verify. Total simulation counts, including those required to train the policy network, should be included. It is also worth discussing the fact that an amortized, trained SBI method can carry out inference on the full posterior without any additional simulator calls. In principle there must be some important scenarios where using local information is more efficient, but this does not seem to have been demonstrated here.

The strength of using local gradients in $\psi$ space over other approaches such as neural density estimation or Bayesian optimization is that this should scale to higher-dimensional parameter spaces. But the highest we see here is 42, well within the reach of existing SBI methods. Furthermore, we never get to see the joint distribution of true and estimated parameters, nor do we get to compare simulation results from true vs. estimated parameters. Therefore, it's hard to say whether the tasks (especially the real world problems) have actually been solved, or whether the current method is simply failing to solve it in a more sample-efficient way than baselines.

It's not so clear why the latent variable $z$ is included, when only the mean output of each surrogate is used to estimate sigma. Do the distributions from individual surrogates show variation, and does this capture uncertainty? Or does including $z$ somehow regularize the surrogates? Do they actually use $z$ or discard it after training?

The section on metrics isn't very clear. Does "objective function" refer to the reward function for RL or the loss for local training? It's not clear in either case how closely either of these correspond to accurate estimation of $\psi$, which is the real aim here. Why not show accuracy of $\psi$ vs. total simulation count? While it's generally true that lowering the objective function means we've gotten closer to the distribution of observables under the ground truth parameters, exactly how close is difficult to determine from the reported metrics. Also, it's not clear to me whether AMO/ANC are calculated over the whole training process, or for a new episode when the policy has already been learned.

Overall the precise sequence of steps is not clearly stated in the main text, and inclusion of a simplified algorithm in the main text would be welcome. Giving a concrete example of what the surrogate training loss might be early on in the manuscript could help readability.

Some relevant citations are missing. Papamakarios and Murray should be cited for SBI. Nonnenmacher (2021) should be cited for emulator-based gradient estimation. Lueckmann (2019) should be cited for simulation-based inference with active learning and ensemble predictions.

References

Lueckmann, Jan-Matthis, et al. "Likelihood-free inference with emulator networks." Symposium on Advances in Approximate Bayesian Inference. PMLR, 2019.

Lueckmann JM, Boelts J, Greenberg D, Goncalves P, Macke J. Benchmarking simulation-based inference. InInternational conference on artificial intelligence and statistics 2021 Mar 18 (pp. 343-351). PMLR.

Nonnenmacher M, Greenberg DS. Deep emulators for differentiation, forecasting, and parametrization in Earth science simulators. Journal of Advances in Modeling Earth Systems. 2021 Jun;13(7):e2021MS002554.

Tejero-Cantero A, Boelts J, Deistler M, Lueckmann JM, Durkan C, Gonçalves PJ, Greenberg DS, Macke JH. SBI--A toolkit for simulation-based inference. arXiv preprint arXiv:2007.09114. 2020 Jul 17.

Papamakarios G, Murray I. Fast ε-free inference of simulation models with bayesian conditional density estimation. Advances in neural information processing systems. 2016;29.

**Questions:**

The decision to train on only local samples makes sense since one only requires local gradients, but there may be common features of the simulator's input-output relations across parameter space. Would it make sense to somehow incorporate older samples as well? Are there obvious cases where one or the other strategy would be preferred? How should one set/explore the size of the memory buffer when applying this approach to a new task?

The study seems to use the same policy network throughout, so we're assuming that the simulator's input/output mapping changes quickly, but the conditions under which we should trust our surrogates changes more slowly? Perhaps this contrast could be discussed in the context of the assumptions we should make about the simulator's local and global properties in order for the proposed method to work well.

Would it make sense to provide the observations $y$ to the policy network?

What are the failure modes? If I try this method out on a new task and the policy or surrogate training fails to adequately learn, how can this be identified? What if I don't know what loss value can be achieved/expected in my system a priori?

Is there any reason to use only 2-layer networks, especially for the 42-dimensional parameter estimation task?

---

### Note · Authors · 2024-11-19

I have read and agree with the venue's withdrawal policy on behalf of myself and my co-authors.